# Evaluating acceptability of the Inpatient Mental Health Pharmaceutical Assessment and Care Tool (IMPACT): A multi-site study in the United Kingdom

Fatima Q. Alshaikhmubarak[1]*, Richard N. Keers[1,2,3], Petra Brown[1,3], Penny J. Lewis[1,2,4]

1 Division of Pharmacy and Optometry, The University of Manchester, Manchester, United Kingdom,
2 NIHR Greater Manchester Patient Safety Research Collaboration, Manchester, United Kingdom,
3 Optimising Outcomes with Medicines (OptiMed) Research Unit, Pennine Care NHS Foundation Trust, Manchester, United Kingdom, 4 Manchester University NHS Foundation Trust, Manchester, United Kingdom

* fatima.alshaikhmubarak@yahoo.com

## Abstract

### Background

The Inpatient Mental Health Pharmaceutical Assessment and Care Tool (IMPACT) was developed to assist pharmacy teams in identifying high risk patients for early intervention. Evaluation of the IMPACT tool is important to ensure its feasibility and effectiveness. This study reports the first evaluation of the IMPACT tool aiming to explore its acceptability by mental health inpatient pharmacy teams using an iterative qualitative approach.

### Methods

Between October 2024 and February 2025, pharmacy staff from five National Health Service (NHS) organisations retrospectively applied the IMPACT tool on patients that they had provided pharmaceutical care to, completed a reflection sheet, and attended an online focus group. Training was delivered to participants before initiating the study and the Theoretical Framework of Acceptability guided the content and analysis of the focus groups.

### Results

Four focus groups and one dual interview were conducted with 12 pharmacists and 5 pharmacy technicians. The tool was viewed as self-explanatory and effective. Most participants were confident using the tool, though some pharmacy technicians reported difficulties due to clinical criteria (e.g., blood tests interpretation) that was not part of their usual duties. Following the first focus group, some changes were made such as clarifying or combining some risk-indicators. These changes were

**Data availability statement:** All relevant data are within the paper and its Supporting information files. Additional data can be requested from The University of Manchester Ethics Committee (contact via research.ethics@manchester.ac.uk) for researchers who meet the criteria for access to confidential data.

**Funding:** This study was part of a PhD funded by the Saudi Arabian Cultural Bureau in London. SE-85881.

**Competing interests:** The authors have declared that no competing interests exist.

well-received by subsequent participants and recommendations and insights gained from all participants assisted in improving the tool.

## Conclusion

This study revealed that the IMPACT tool was acceptable by pharmacy team members and resulted in a refined version. Future work should further explore the tool's feasibility and impact using mixed methods approaches.

## Background

Traditionally, clinical pharmacists approached inpatients for medication review with no specific criteria to target those of higher risk for medication harm. However, this approach has been recently questioned as it may mean that a patient with high-risk of medication harm could be overlooked while pharmaceutical care is being directed toward lower risk patients [1]. Recognising the need for a systematic approach to guide them in reviewing their patients and improving workflow, pharmacists around the world started developing and using patient prioritisation tools [2–8]. Studies reported that patient prioritisation approaches not only ensure timely review of patients reducing their risk of medication harm but also support pharmacy team members to manage their workload and work more efficiently [4,5]. Despite the publication of several patient prioritisation tools for pharmacy services [2–8], and reviews of published pharmaceutical prioritisation tools [2,4,6–8], none have reported a prioritisation tool specifically developed for mental health wards. A recent study explored pharmaceutical patient prioritisation approaches within UK mental health wards reported considerable interest in patient prioritisation, with 21 National Health Service (NHS) trusts and boards utilising patient prioritisation approaches [9]. It also highlighted an absence of a standardised patient prioritisation tool for adoption and use by mental health pharmacy teams. Existing prioritisation approaches have largely been developed for a general acute care context making them difficult to use in mental health wards where patients have different risk factors for medication harm [10]. In addition, service delivery, patient characteristics, and medication use differ in mental health wards necessitating different patient prioritisation criteria [9]. This inspired the creation of the Inpatient Mental Health Pharmaceutical Assessment and Care Tool (IMPACT), an evidence-based tool developed specifically for UK inpatient mental health pharmacy teams to prioritise patients for pharmacy review [11]. The IMPACT tool was developed using Delphi questionnaires informed by a systematic review [10] that identified risk factors for drug related problems within mental health wards and a multi-method study [9] that explored existing prioritisation approaches used by mental health pharmacy teams in the UK. A flowchart charting the IMPACT tool development process can be seen in Fig 1.

Whilst it is hoped that the IMPACT tool, being rigorously developed, would greatly benefit mental health pharmacy teams and improve patient outcomes, assessing its usefulness and feasibility is essential. Feasibility testing is necessary to identify

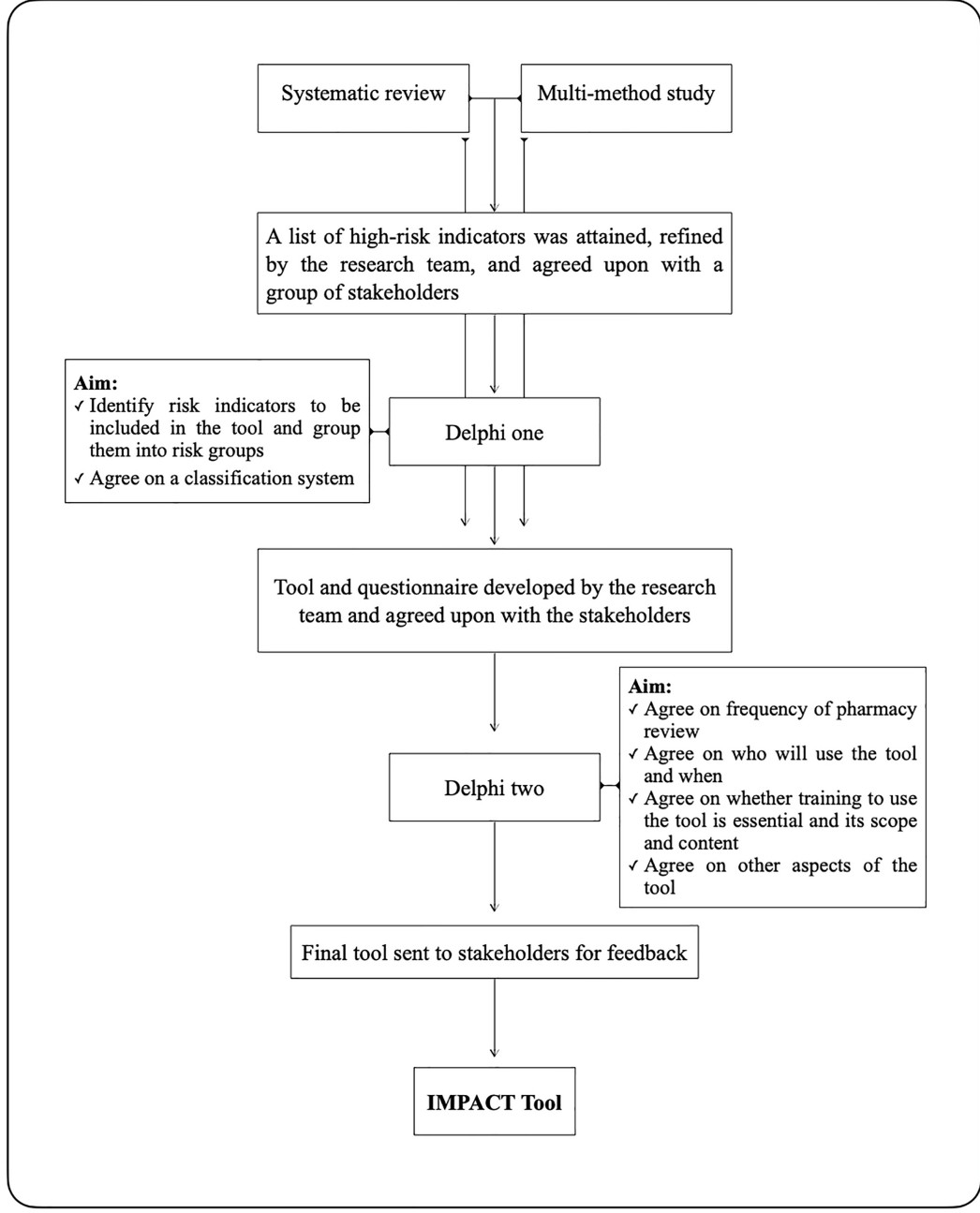

**Fig 1. A flowchart of the IMPACT tool development process [11].**

areas for improvement before wider adoption and implementation and to determine whether an intervention is suitable for further assessment [12]. Klaic et al recommended in their framework of implementability of healthcare interventions that acceptability testing should be the initial step prior to implementation and further evaluations of new interventions [13]. They reasoned that low acceptability suggests low fidelity and feasibility and hence further evaluations prior to acceptability would be a waste of efforts and resources [13]. Considering the crucial role of acceptability on uptake, adoption, and

sustainability of interventions [14–16], this study aimed to test the acceptability of the IMPACT patient prioritisation tool by UK mental health pharmacy team members.

## Methods

### Study design

The study involved sequential events including a training session, a time period for using the tool and completing reflection sheets, and online focus groups with pharmacy staff members. An iterative approach was used where the tool was modified following the first focus group and the modified version was tested with subsequent participants. A flowchart explaining the sequence of events in this study can be seen in Fig 2.

### IMPACT tool

The IMPACT tool was designed for use by mental health pharmacists and pharmacy technicians to categorise patient into three risk groups (red = high-risk, amber = medium-risk, green = low-risk) for pharmacy review [11]. Pharmacy team members are expected to use the IMPACT tool during or after medication reconciliation by reading a list of risk indicators, ticking the boxes of indicators that apply to the patient, and finally assigning a risk category based on identified risk indicators [11]. The IMPACT tool (version 1) can be viewed in S1 File.

### Sample and recruitment

Pharmacists and pharmacy technicians with different levels of experience working within mental health inpatient wards from five NHS trusts and boards were invited to take part in this study. Included NHS organisations were identified through (a) the research teams' network, (b) chief/senior pharmacists who participated in previous development work and were

**Flowchart of study events**

Advertise the study at a pharmacy team meeting and/or through distributing invitation emails to pharmacy staff at participating NHS organisations.

Interested individuals email the research team to express their interest in the study.

An online training session is organised with participants.

Participants complete the patient prioritisation tool for around 10 patients and use the reflection sheets to document their views and opinions.

Reflection sheets are collected and participants attend a focus group to discuss their views on the tool.

**Fig 2. A flowchart of study events.**

interested in the IMPACT tool [9], and (c) through presentation of the IMPACT tool at the College of Mental Health Pharmacy conference 2024. A brief description of each participating organisation is presented in Table 1 and detailed information are presented in S2 File.

An email invitation was distributed to pharmacy team members at participating organisations. Interested staff who emailed the research team were subsequently sent the participant information sheet and consent form to consider participation. At two organisations, this was preceded by a presentation to pharmacy team members via Microsoft Teams (MS Teams) to introduce the concept of patient prioritisation, development of the IMPACT tool and study and describe what participation would involve. Recruitment commenced on 25th July 2024 and concluded on 24th January 2025 and all participants provided written informed consent. Participants were offered a £25 high street voucher as compensation for their time after attending the focus groups.

### Training session

Participants were invited to attend an online training session aiming to introduce the research, present the tool and describe how it should be used, and explain the steps required for the current study. The training session materials were slightly modified following the first focus group and, as an iterative process was utilised, all changes to the tool based on the first focus group were explained for the subsequent participants.

### Data collection

Following the training session, participants completed the following study stages:

**Stage 1:** each participant completed the IMPACT tool for approximately 10 newly admitted or existing patients over a period of up to 3–4 weeks. Participants were given the option to either print the tool or view it electronically and it was emphasised that patient care should not change or be influenced by the tool's outcome.

**Stage 2:** each participant completed a reflection sheet designed to help participants keep a record of the patients they completed the tool for and note their agreement/disagreement with the tool outcome with justification of their decision.

**Stage 3:** each participant attended one online focus group via MS Teams moderated by FQA to discuss their experience using the tool. The guide for the focus groups (S3 File) was developed using the Theoretical Framework of Acceptability (TFA) [17]. All focus groups were recorded and automatically transcribed via MS Teams and transcriptions were checked for accuracy by FQA.

Table 1. A brief description of participating NHS organisations.

|  | NHS Borders | LSCFT | OHFT | PCFT | SWYT |
|---|---|---|---|---|---|
| Pharmacy team size | 2 pharmacists 1 pharmacy Technician | 57 pharmacists 34 pharmacy technicians | 18 pharmacists 10 pharmacy technicians | 23 pharmacists 10 pharmacy technicians | 18 pharmacists 18 pharmacy technicians |
| Number of inpatient beds | 45 beds | 709 beds | 407 beds | 486 | 528 |
| Wards distribution and geographical locations | 4 wards across 2 sites | 50 wards across 23 sites | 25 wards across 7 sites | 33 + inpatient wards across 7 sites | 35 wards across 5 hospital sites and three off site rehab units |
| EPMA | Not implemented | Fully implemented | Fully implemented | Implemented on one ward only | Fully implemented. |
| Dispensary | No dedicated mental health dispensary | No dedicated mental health dispensary (except one site) | In house dispensary | No dedicated mental health dispensary | In house dispensary (except two sites) |

LSCFT: Lancashire and South Cumbria NHS Foundation Trust, OHFT: Oxford Health NHS Foundation Trust PCFT: Pennine Care NHS Foundation Trust, SWYT: South West Yorkshire Partnership NHS Foundation Trust. EPMA: Electronic Prescribing and Medicines Administration.

The TFA describes seven components or constructs to assess acceptability (affective attitude, burden, perceived effectiveness, ethicality, intervention coherence, opportunity costs, and self-efficacy) which are presented in Table 2 [17]. Aside from being commonly used, it was designed specifically for acceptability testing of healthcare interventions and it includes assessing the acceptability of intervention content which may help improve the intervention at the prospective (anticipated) acceptability stage and prior to implementation making it a suitable model to test the IMPACT tool.

## Data analysis

The transcripts were coded and thematically analysed using NVivo software following Braun and Clarke's guide [18] incorporating a mix of inductive and deductive approaches guided by the TFA model. Initially, the first focus group transcript was analysed manually to identify areas for improvement of the IMPACT tool and training material. All focus groups were then coded using NVivo software and themes were identified under each construct of the TFA model. Potential alterations to the tool and training material were identified from codes under each TFA construct. As data collection progressed, fewer new themes emerged across successive focus groups and the dual interview, with no significant new themes identified in the final data collection, suggesting that data saturation had been achieved. The completed reflection sheets were summarised and suggestions or ideas for improvement were extracted to augment the focus group findings.

## Ethical approval

Ethical approval was obtained from the University of Manchester Ethics Committee (19507) and from the Health Research Authority including HRA and HCRW Approval for the trusts in England and NHS/HSC R&D Permission for the health board in Scotland (IRAS: 341745).

## Results

A total of 18 pharmacy staff members consented to participate in this study. One pharmacist dropped out from the study as they struggled to find time to participate. Consequently, a total of four focus groups and one dual interview were conducted with twelve pharmacists (n = 12/17, 70.6%) and five pharmacy technicians (n = 5/17, 29.4%) between October 2024 and February 2025. One focus group lacked pharmacy technician representation, while the dual interview lacked a pharmacist's contribution. The duration of training sessions varied between 14–44 minutes depending on factors such as the number of attendees, the questions asked, and the time required to schedule the focus group.

Participants had 3–4 weeks to use the tool prior to the focus group except for three participants who had 1–2 weeks due to time constraints. Participants had variable experience within mental health pharmacy ranging between one year

**Table 2. Definitions of the seven TFA constructs as described by Sekhon et al. [17].**

| TFA constructs | Definition |
| --- | --- |
| Affective Attitude | How an individual feels about the intervention |
| Burden | The perceived amount of effort that is required to participate in the intervention |
| Ethicality | The extent to which the intervention has good fit with an individual's value system |
| Intervention Coherence | The extent to which the participant understands the intervention and how it works |
| Opportunity Costs | The extent to which benefits, profits or values must be given up to engage in the intervention |
| Perceived Effectiveness | The extent to which the intervention is perceived as likely to achieve its purpose |
| Self-Efficacy | The participant's confidence that they can perform the behaviours required to participate in the intervention |

to over 35 years. They worked within diverse settings including acute adult wards, dementia, forensics, rehabilitation and high support, learning disabilities, eating disorders, child and adolescents, and psychiatric intensive care and six reported having managerial duties at the time. Focus groups lasted between 44–89 minutes (average 64.4 minutes).

### Iteration rounds

An illustration of the iterative development process can be seen in Fig 3. The first focus group was conducted with three pharmacists and one pharmacy technician. Changes to version 1 of the tool were then discussed and agreed upon with the research team before being tested with subsequent participants. The agreed changes to version 1 of the tool are

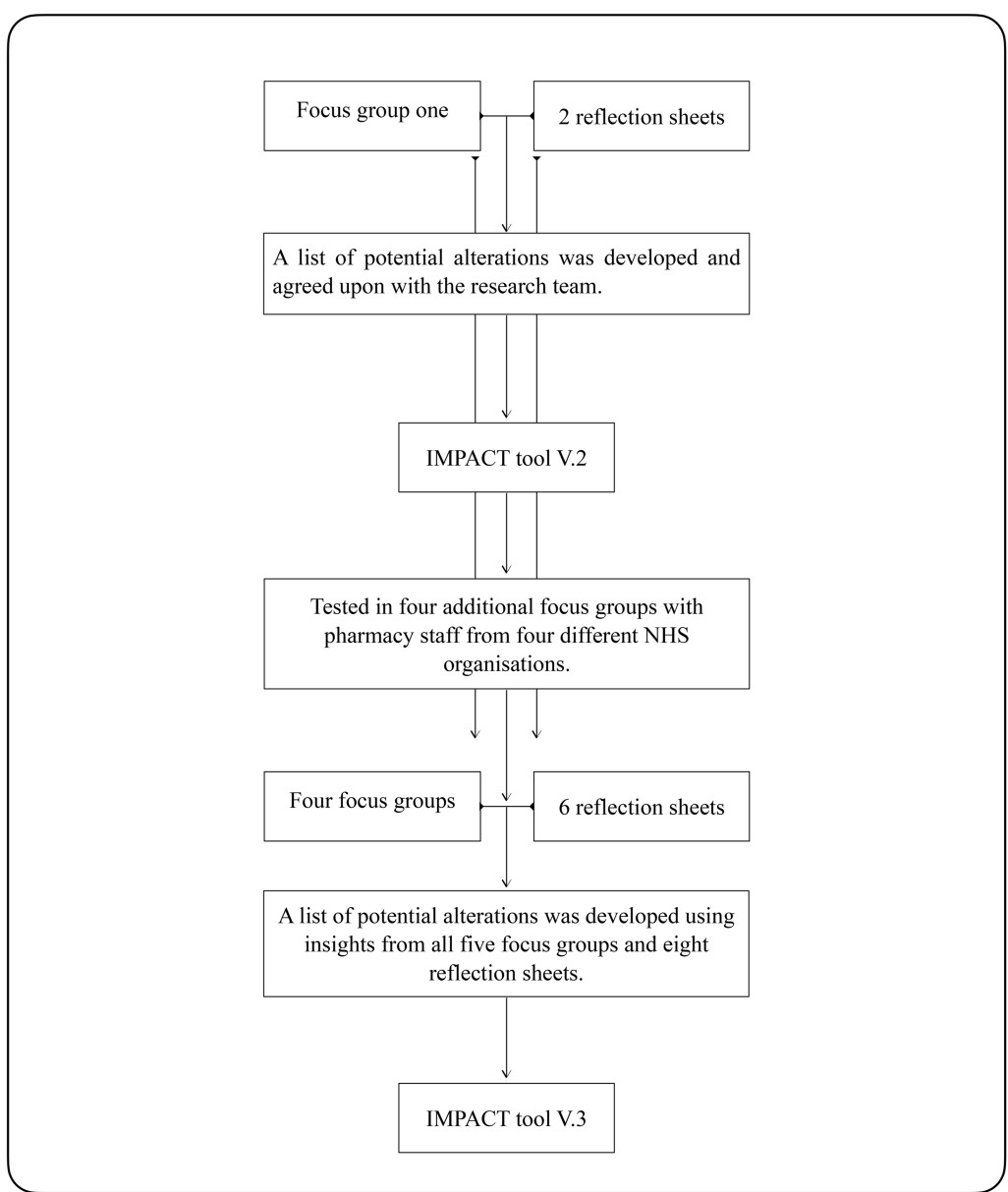

**Fig 3. A flowchart of the iterative development of the IMPACT tool.**

presented in Table 3 and the changes to the training material are presented in Table 4. The modified tool (version 2) with highlighted changes and the updated training session material can be seen in S4 File. All the changes introduced to the tool following the first focus group were acceptable by subsequent participants and remained the same in the final version (version 3) of the tool (Table 3).

## Reflection sheets

Only eight participants (six pharmacists and two pharmacy technicians) from three organisations provided their reflection sheets, with time limitations reported as the main reason for non-completion. Three participants, being from a single organisation, collectively completed the tool for 10 patients.

Of the completed reflection sheets, participants agreed with the tool's outcome for 67.5% of the patients (n = 37/54), disagreed with the tool's outcome for 18.5% (n = 10/54), and neither agreed nor disagreed for 13% (n = 7/54). Only one

**Table 3. Modifications to the IMPACT tool version 1.**

| Modifications following the first focus group | |
| --- | --- |
| **Original risk indicator** | **Modified risk indicator** |
| Patient <u>requires</u> intramuscular rapid-tranquillisation administration | Patient <u>administered</u> intramuscular rapid-tranquillisation administration |
| Patients with physical healthcare issues requiring follow-up | Patients with physical healthcare issues requiring follow-up <u>by pharmacy team</u># |
| Female of child bearing potential prescribed teratogenic medicines such as sodium valproate | Female <u>or male</u> of child bearing potential prescribed teratogenic medicines such as sodium valproate <u>and does not have relevant authorisation and pregnancy prevention plan</u> |
| Anticonvulsants (e.g., topiramate, levetiracetam) | Anticonvulsants (e.g., topiramate, levetiracetam) *<u>for seizure</u> |
| **Red criteria:**<br>High Risk Medicines<br>High Dose Antipsychotic Therapy (above 100% BNF maximum) prescribed | Red criteria:<br>Newly prescribed or changed (e.g., dose change) high risk medicine<br>Amber criteria:<br>Patient taking high risk medicines (Refer to the high risk medicines table on the right)<br>*High Dose Antipsychotic Therapy was added to the high risk medicines list. |
| Fall ≥ 1 in the preceding 3 months | History of fall (For electronic tool: Fall ≥ 1 in the preceding 3 months) |
| **Combined risk indicators** | |
| **Original risk indicator** | **Combined risk indicators** |
| • Chronic kidney Disease Stage ≤ 3a (eGFR ≥ 45 ml/min)<br>• Electrolytes levels outside reference range | Chronic kidney Disease Stage ≤ 3a (eGFR ≥ 45 ml/min) <u>OR</u> electrolytes levels outside reference range |
| • Missed >2 doses of the same regular prescribed medication<br>• Patients with non-adherence<br>• Patients regularly spitting out or refusing medication | Patient missed >2 doses of the same regular prescribed medication (e.g., due to non-adherence) <u>OR</u> regularly spitting out or refusing medication |
| • White blood cells (WBC) outside reference range<br>• Haemoglobin levels (CRP, HB1) outside reference range | White blood cells (WBC) <u>OR</u> Haemoglobin levels (CRP, HB1) outside reference range |

#Participants suggested this on the basis that the indicator should only flag risk when timely pharmacy input is required.

**Table 4. Modifications to the training material accompanying version 1.**

**Changes to the training material**

- The example of using the tool was expanded by including simulated patient information.
- The following considerations were added:
  - The tool is intended to be used on admission (during or after medication reconciliation) and then reviewed every time the patient is seen.
  - The tool should be used flexibly (e.g., blood tests may not be available but it is there for you to complete it when you can)
  - The frequency of review is just for guidance, and the tool might help you spend more time with red patients rather than reviewing all patients quickly.
  - This tool was primarily developed for use in adult wards so some criteria may not be applicable, future work may explore speciality specific tools.
  - It expected that professional judgement is used when necessary and that 'other' allows for the professional to escalate or deescalate patients if required- the tool cannot cover every patient presentation and risk.
  - This evaluation is key in improving the tool and making sure it is practical to use.

participant used the 'other' tick box (which allows escalating or de-escalating patients when clinically appropriate) whereas others reported recording patient cases requiring the use of the 'other' tick box as disagreements. Additionally, three disagreements originating from version 1 of the tool were resolved after modifying the tool. A summary of the reflection sheet results can be seen in Table 5.

## Focus groups

Key themes derived from the analysis of focus group data are presented below under each construct of the TFA framework.

**Affective attitude.** Most participants felt comfortable using the IMPACT tool and all participants had positive views towards it. Key contributors to this positive view included the clarity of the tool making it easy to understand, the use of colour coding and tick boxes, and relevance of the risk indicators included. For example, several participants appreciated the content of the tool as a helpful reminder of key risk indicators that may sometimes be overlooked.

**Table 5. Reflection sheets summary.**

| # | No. of patients | Patients hospitalised>7 days | Outcome | Agreement with the tool |
|---|---|---|---|---|
| 1 | 10 | 9 | All red | Agreement: 8 patients<br>Disagreement: 2 patients |
| 2 | 10 | 7 | 5 red<br>4 amber<br>1 green | Agreement: 9 patients<br>Disagreement: 1 patient |
| 3 | 10 | 6 | 5 red<br>5 amber | Agreement: 8 patients<br>Disagreement: 2 patients |
| 4,5,6 | 10 | 1 | 7 red<br>3 amber | Agreement: 4 patients<br>Disagreement: 4 patients<br>Neither: 2 patients |
| 7 | 9 | – | 3 red<br>6 amber | Agreement: 5 patients<br>Disagreement: 1 patient<br>Neither: 3 patients |
| 8 | 5 | 3 | 3 red<br>2 amber | Agreement: 3 patients<br>Neither: 2 patients |

*"I did kind of like having that [list of risk indicators] all in front of me and so that made it … like a good experience to complete." (P8, pharmacist)*

Despite this overall positive affective attitude, four pharmacy technicians expressed some discomfort completing the tool. This was chiefly due to the difficulties they faced completing the tool as some risk indicators were not considered part of their usual duties and they felt that they would require some further training to use the tool effectively.

*"So this is quite clinical. So I was a bit like totally out of my comfort zone doing this and it's different to like any other work I usually do. But it was like a nice change I suppose, but it didn't come naturally." (P12, pharmacy technician)*

This view was not shared by another pharmacy technician from a different organisation who felt comfortable completing the tool as they considered that it aligned well with their daily work.

**Burden.** All pharmacists found the tool easy to use and self-explanatory, a view shared by two pharmacy technicians. The other pharmacy technicians found the tool difficult to use, mainly due to clinical factors that they reported were not usually part of their routine work such as identifying reference ranges for some risk indicators (e.g., creatine kinase).

A commonly discussed barrier to using the tool was the length of the time needed to complete it. However, it was also highlighted that reviewing patients takes time regardless of the tool use, and the tool serves to formalise the process.

*"I'm not sure if kind of like making it [the time completing the tool] faster should be a priority because we want to spend time and do a thorough assessment on our patients, so I think it really needs the time that it needs and it's really depending on the patient." (P15, pharmacist)*

Formatting of the tool was another issue as participants that printed the tool and used it in paper format struggled with readability due to the small text size and found the use of paper impractical. Participants also believed that combining or reorganising some risk indicators may reduce the burden of completing the tool (see combined risk indicators in Tables 3 and 6).

The third barrier to using the tool was perceived ambiguity and subjectivity of some risk indicators. For example, one participant described difficulties in making a clinical judgement regarding self-harm and suicidal thoughts. However, when presented with the suggestion to make them more explicit, the same participant cautioned that doing so might limit flexibility reducing the tool's uptake.

*"… mental health is very complex field and for reasons like everyone's different, everyone's unique, everyone's got different backgrounds … I just think you're better off leaving it [subjective risk indicators] open because if the tool is too objective, too closed, it would potentially discourage certain people from using it." (P5, pharmacist)*

A participant from a different focus group viewed ambiguity positively, noting that most risk indicators require some degree of clinical judgment, with only a few being explicitly defined (e.g., age).

All participants believed that training could help improve the usability of the tool. For example, training on how to use the tool and where to find the information needed to complete it. Other suggestions proposed included that pharmacy technicians complete the parts of the tool that are relevant to their work then pass it to a pharmacist to complete the remaining parts or to have a simplified version of the tool (e.g., including reference ranges) tailored for pharmacy technicians.

Familiarity was believed to make the use of the tool easier and quicker as individuals become accustomed to using it, memorising its content. Consistent use of the IMPACT tool by all staff was thought to facilitate the handover of patient risk classification improving ward coverage (i.e., managing patient-related tasks in alternative ward locations during staff absences).

**Table 6. Modifications to the IMPACT tool version 2.**

**Tool modifications based on acceptability testing**

| Original risk indicator | Modified risk indicator |
|---|---|
| Newly prescribed or changed (e.g., dose change) high risk medicine | High-risk medicines that are newly prescribed, changed (e.g., dose change), or <u>exhibit irregularities in serum levels</u> |
| Patient <u>requires</u> oral 'when required' psychotropic for agitation | Patient administered oral 'when required' psychotropic for agitation |
| Prescribed more than one hypnotic, anxiolytic, or antidepressant | Prescribed >1 hypnotic OR >1 anxiolytic OR >1 antidepressant (this includes both PRN and regular prescriptions) |
| Patients planned for discharge/leave | Patients planned for discharge/leave with outstanding issues requiring follow up |

Other changes

- Physical health issues were combined in a supplementary box & a single statement was added in the risk indicators list: "Physical health issues (refer to the red/amber box on the right)"
- The format of the tool was updated to increase readability (e.g., alternating shading).
- A reminder was added below the amber risk indicators: "Reminder: identifying ≥4 amber criteria for an individual patient will escalate them to a 'red' risk rating"

Combined risk indicators

| Original risk indicator | Combined risk indicators |
|---|---|
| • Age <18<br>• Age >70 | Age (<18 OR >70) |
| • Formulation review required, e.g., NG, PEG, JEJ<br>• Patients with swallowing difficulties/ Nil by mouth<br>• Patients receiving covert medications | Formulation review required (e.g., due to swallowing difficulty or covert administration) |
| • New T2/T3<br>• T2/T3 renewal needed | New T2/T3 or renewal needed |

*"You know you get used to where things are and things that you need to consider. And as I say, eventually, in theory, the piece of paper wouldn't necessarily be needed … but you've got it there so that you could then hand it on to a colleague if need be." (P1, pharmacist)*

In contrast, participants from another organisation believed that familiarity would not change the time needed for tool completion as the process required extensive searching through patient files, which was thought to be time-consuming regardless of familiarity.

*"The tool won't sort of change the level of what you need to look into regardless [of consistent use], really. So I think it would take a similar amount of time" (P7, pharmacist)*

Lastly, digitalisation of the tool was advocated by all participants to streamline the patient prioritisation process and optimise documentation. Fig 4 presents the reported barriers and facilitators for using the IMPACT tool.

**Ethicality.** Most participants believed the IMPACT tool to be fair because pharmacy staff were considered to have limited time, and the tool allowed each patient to be assessed using the same approach rather than relying solely on clinical judgement.

*"I think it is fair because you are dealing with each individual in exactly the same way, you are assessing them in the same way, so there should be no bias in relation to how you would assess that individual." (P1, pharmacist)*

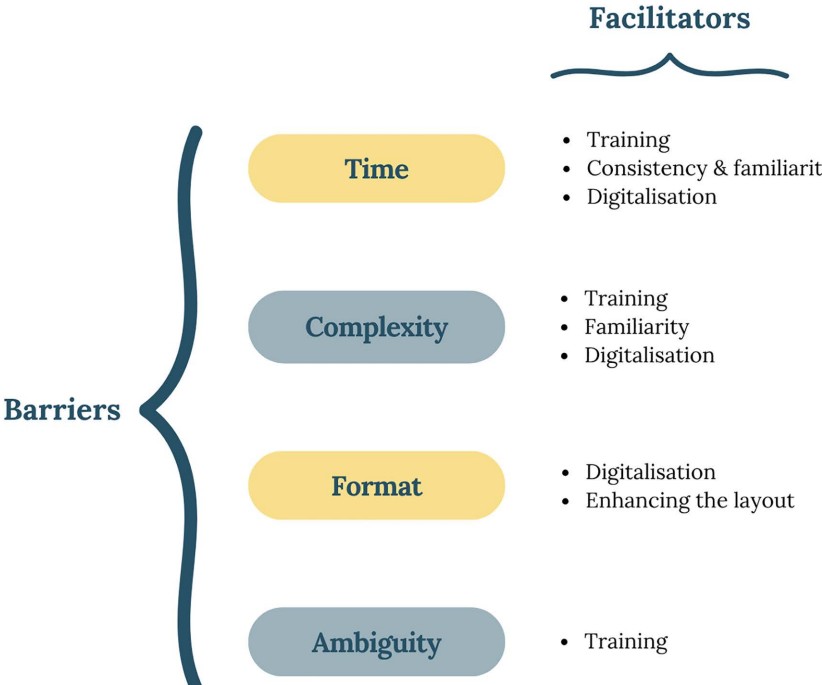

**Fig 4. Barriers and facilitators perceived to influence the burden of using the IMPACT tool.**

One participant expressed that using a prioritisation tool may be considered fairer than not using a tool and that patients may feel assured knowing that a standardised tool is used.

*"In some ways, it's not really defensible not to use something like this … I can imagine people will feel comforted that there is something structured." (P3, pharmacist)*

One ethical concern raised with using the IMPACT tool was that the lower-risk 'green' rated patients might receive less attention from the pharmacy teams.

**Intervention coherence.** The concept of a patient prioritisation tool was generally well understood and valued by participants. However, one participant questioned the need for using the tool if they could review patients daily.

*"I guess what it would mean for me is that there would be some patients on the ward that I would review less frequently based on the tool, but there are some things that you notice by reviewing all patients every day … Would it be quicker just to review all patients rather than use the tool and not review some patients?" (P2, pharmacist)*

When asked if there were days they couldn't review all patients, they reflected that it happens although not frequently, but the tool would be useful in such situations. They also thought the tool might help allocate more time to higher risk patients.

*"That's a good valid point that it might be helpful in those scenarios when maybe we don't have sufficient time to review all patients. I guess it would also mean that we could maybe target more time to the patients that require it" (P2, pharmacist)*

Participants raised concerns about the usability of the tool across different sites. For example, one participant explained that high-risk patients are usually moved to another site that is more equipped to support patients requiring closer monitoring. Hence, the patients categorised as high-risk within their site might be considered medium-risk patients at other sites.

*"A really high-risk patient for us that needs that intense input goes to a completely different [site]. I suppose, and that's just the nature of how our service is structured. So that's what I mean by like tailoring of the risk to the area in terms of what's red to us maybe means twice a week reviews, amber's once a week, green is like periodic" (P8, pharmacist)*

Participants also suggested tailoring the tool to different specialties. Some believed the current tool may be useful for acute wards but less useful in a psychiatric intensive care units where almost all patients would be categorised as red, with the current criteria, or in a rehabilitation ward where the frequency of review depends on the frequency of pharmacy ward visits.

*"If you're looking after a psychiatric intensive care unit … there is a high acuity. There's a lot of input from clinical pharmacy involvement in the MDTs [multidisciplinary teams] on a daily or frequent basis … so interestingly, the tool may be less beneficial in that setting then" (P1, pharmacist)*

The differences between the roles of pharmacy staff were also seen across different sites within the same organisation. For example, while one participant stated that reviewing whether a patient had an echocardiogram (ECG) was not their responsibility, another replied that they often intervene to convince patients to have an ECG done when they refuse.

*"I think the experience just differs from pharmacist to pharmacist because I'm quite involved with ECGs, like when the doctors can't convince patients to get an ECG, I kind of go on the ward with them and I'm kind of like alright, you're going to have an ECG, and the patient suddenly goes from I don't want an ECG to where's my ECG?" (P5, pharmacist)*

Participants did not always have a clear understanding of the main purpose of the risk indicators and sometimes questioned the presence of non-modifiable factors (e.g., age), factors that may not frequently change (e.g., substance abuse), and factors that in their view may not require a pharmacy intervention (e.g., outstanding ECG).

*"I mean, the thing with ECG, they might be refusing it. You know, so it's just that nuance because we're mental health there's often a lot of refusals going on, at least on my ward. And in that sense, it wouldn't necessarily be a trigger to keep seeing them every couple of days because it's not going to happen" (P3, pharmacist)*

When the purpose of the risk indicators was clarified -which is to categorise patients into risk groups to prioritise their review rather than requiring pharmacy staff to address individual risk indicators- one participant suggested that this intention would need to be made clearer to users.

*"…maybe it's just partly a change of mindset as well. Like for us, you know that holistically they are still high risk because we have all of these risk factors, and even if we're not necessarily doing anything about them every day, they're still high risk because of those factors, so yeah." (P9, pharmacist)*

Some participants highlighted benefits of the tool such as benchmarking, accountability, standardisation of care, allocation of resources, and to facilitate handover. They reported it would be particularly useful for new admissions to ensure patients received the baseline pharmacy review they needed.

**Opportunity costs.**  Participants reported highly variable durations for completing the tool for individual patients ranging from less than five minutes up to 40 minutes. Pharmacy technicians who expressed difficulties using the tool generally reported taking longer than most to complete it. Participants reported that the time to complete the tool varied greatly depending on factors not related to the tool such as having prior knowledge of the patient and availability of information.

While the reported time to complete the tool varied, participants consistently viewed the time spent as a worthwhile investment.

*"I think you know if it frees our clinical time to, you know, focus on those real high-risk patients, then yeah, it is [worth the time spent completing it]… I think it would be a good thing in practice." (P11, pharmacy technician)*

Several participants noted that they had their own list of risk indicators, either documented or in their mind, that they used to review and prioritise patients. The use of the IMPACT tool added to their internal list, standardising this process, and formalising it. Hence, the use of the tool was not viewed by these individuals as adding a substantial time burden.

*"To be fair, I would be doing this anyway, as [name] said. Before there's a checklist in your head. So I would be doing this anyway, so it's not like it's taken me more [time]" (P3, pharmacist)*

**Perceived effectiveness.**  The IMPACT tool was generally well received by participants who thought the tool represented medicines optimisation and safety issues affecting inpatients. There was also considerable agreement with the tool outcomes as supported by the analysis of the reflection sheets.

Nevertheless, some participants questioned the ability of the tool to accurately categorise patients into risk groups. Whilst many of them believed that using their clinical judgement when disagreeing with the tool's outcome (by ticking the 'other' option in the tool) would likely solve this problem, one expressed a concern that overreliance on clinical judgement might reduce the usefulness of the tool.

*"I think the only risk with that [interviewer] is that then we end up overriding so many that kind of, you know, reduces the confidence, or you know, accuracy of the tool if you like" (P3, pharmacist)*

Most participants believed the IMPACT tool would improve patient outcomes and pharmacy service delivery with some suggesting formal evaluation for a clearer demonstration of its effectiveness. Some participants suggested that the tool would be particularly useful for newly qualified pharmacists or individuals new to mental health settings.

**Self-efficacy.**  The majority of participants felt confident using the IMPACT tool due to its ease of use and perceived benefits. Several participants, however, suggested that more practice using the tool may increase their confidence.

Some pharmacy technicians were not fully confident they could use the tool effectively in the context of their current workload. This was mainly because to them it was felt difficult to complete and time consuming.

*"On a Monday we get up to like eight new admissions … it's just me and a pharmacist and it's like if I was told to do this for eight people, I probably wouldn't get to do it right until the end of week." (P12, pharmacy technician)*

**Overall acceptability.**  Participants reported the IMPACT tool to be generally acceptable. Some thought it could be used as it was, whereas others thought it would be easier to use once it was modified based on their suggestions, implemented and further improved, or incorporated into an electronic system. Additionally, participants discussed the practical challenges of implementing the tool. For example, they suggested that successful adoption may require mandating its use to ensure consistent application.

## Final changes to the IMPACT tool

Participants' feedback and suggestions to improve the IMPACT tool were extracted from the five focus groups, supported by the eight reflection sheets, and carefully considered by the researchers. Following consideration, it was believed that a usage manual to accompany the IMPACT tool would be useful to assist organisations in implementing and using the tool. The training material was expanded and incorporated within the usage manual with recommendations for adapting and updating it locally. The usage manual (including the training material) and the final version of the IMPACT tool (version 3) can be seen in S5 File and Fig 5 respectively. To offer flexibility, version 3b was also developed where the instructions and supplementary boxes were placed on a separate page, allowing for larger text size and double-sided printing (S6 File). In addition, a list of changes to the IMPACT tool version 2 (to achieve version 3) are presented in Table 6 and examples of the considerations included in the usage manual are described in Table 7.

## Discussion

This study has revealed that the IMPACT patient prioritisation tool was acceptable and well received by participants. It also revealed a 67.5% agreement and 18.5% disagreement between the tool's prioritisation and the participants' prioritisation based on their own clinical judgement. Participants' feedback was invaluable guiding the necessary refinements to optimise the tool and ensure its alignment with user needs and clinical realities. Changes applied to the IMPACT tool version 1, were well received by subsequent participants. Additional changes applied to version 2 led to the development of version 3 which is more concise and easier to use (Fig 5) and a usage manual (S5 File).

There was general agreement on the risk indicators in the IMPACT tool with minor changes to achieve clarity and specificity. Some suggestions, however, were not applied to the IMPACT tool due to reasons such as lack of consensus and could be further explored in future research. For example, participants consistently suggested modifying the polypharmacy risk indicators by increasing the threshold or excluding certain medications as patients may be taking several medications that may be considered 'low risk' (of safety issues) such as vitamins. Participants were describing problematic polypharmacy, which is a situation "where multiple medications are prescribed inappropriately, or where the intended benefit of the medication is not realised" [19]. Whilst the number of medications may not be the only factor in problematic polypharmacy, it is commonly used to measure polypharmacy due to its simplicity and ease of measurement [19] which is the reason it was used in the IMPACT tool. It is acknowledged, however, that the growth in prescribing may warrant continuous update of the number of medications used to define polypharmacy [19]. Hence, future research could further explore appropriate measurement of polypharmacy and whether increasing the threshold is necessary.

The positive views toward the IMPACT tool by the majority of participants were perhaps expected as stakeholders were involved throughout the tool development process to ensure its acceptability [11]. The identified perceived effectiveness and confidence in using the IMPACT tool by pharmacists may predict an increased likelihood of its adoption [20]. However, there could be some self-selection bias as pharmacists who considered that improvements may be made to the current pharmacy service delivery approach may have been more likely to participate in the study. Therefore, future work should involve testing the tool in a wider context. As pharmacy technicians were generally less confident in using the IMPACT tool, they might be less likely to adopt it, although their confidence might increase when using the newer version of the IMPACT tool and undertaking appropriate training.

The use of the IMPACT tool may contribute to upskilling pharmacy technicians, aligning with UK initiatives to expand pharmacy technicians' clinical roles to enable pharmacists to focus on more advanced clinical duties [21]. Participants in our study suggested that pharmacy technicians may use a simplified version of the tool (e.g., including reference ranges) or complete certain sections before handing it to pharmacists to complete the remainder. Whilst using a different version of a patient prioritisation tool by pharmacy technicians was observed in one organisation in the UK [9] and might work well for some organisations, completing different sections of the tool has not been observed before. As all the identified

 

**Fatima Q. Alshaikhmubarak, Richard N. Keers, Petra Brown, Penny J. Lewis.**
**2025**

**The University of Manchester** — MANCHESTER 1824

## Inpatient Mental Health Pharmaceutical Assessment and Care Tool (IMPACT) Version 3

| Patient name: | | Admission date: |
|---|---|---|
| Patient ID: | | Ward: |

### Red

| | | |
|---|---|---|
| **Drug Related** | ☐ | Presence of significant* drug interaction <br> *an interaction occurred that either requires action to be taken to avoid harm or may have contributed to or resulted in patient harm |
| | ☐ | Presence of significant** adverse drug reaction (ADR) <br> **an adverse drug reaction occurred that may have contributed to or resulted in patient harm |
| | ☐ | Polypharmacy ≥ 10 regular medications |
| | ☐ | Missed doses of high risk medications |
| | ☐ | More than one regular antipsychotic prescribed |
| | ☐ | Sudden/abrupt cessation of medication |
| | ☐ | Patient administered intramuscular rapid-tranquillisation since last IMPACT tool review |
| | ☐ | Patient prescribed medicines as part of a clinical trial |
| | ☐ | High-risk medicines that are newly prescribed, changed (e.g. dose change), or exhibit irregularities in serum levels (refer to the high-risk medicines box on the right) |
| **Patient Related** | ☐ | Patient <12 years (age) |
| | ☐ | Patient with dementia or cognitive impairment prescribed one or more antimuscarinics |
| | ☐ | Formulation review required such as NG, PEG, JEJ (e.g. due to swallowing difficulties/Nil by mouth, or covert administration) |
| | ☐ | Patient receiving electro-convulsive therapy (ECT) |
| | ☐ | Female or male of child bearing potential prescribed teratogenic medicines such as sodium valproate and does not have or needs updating relevant authorisation and/or pregnancy prevention plan |
| | ☐ | Patient has ≥ 4 amber criteria |
| | ☐ | Physical health issues (refer to the red box on the right) |
| | ☐ | Other …………………………………… |

### Amber

| | | |
|---|---|---|
| **Drug Related** | ☐ | Patient taking high risk medicines (refer to the high risk medicines box on the right) |
| | ☐ | Patient missed ≥2 doses of the same regular prescribed medication (e.g. due to non-adherence) OR regularly spitting out or refusing medication |
| | ☐ | Patient with unverified newly started medication |
| | ☐ | Polypharmacy ≥ 5 regular medications |
| | ☐ | Prescribed >1 hypnotic OR >1 anxiolytic OR >1 antidepressant (this includes both PRN and regular prescriptions) |
| | ☐ | Patient prescribed unlicensed medicines |
| | ☐ | Increase of a regular psychotropic within 7 days of the last increase (unless as part of a dose titration regimen) |
| | ☐ | Any single drug above BNF limits (unless planned detoxification) |
| | ☐ | Patient administered oral 'when required' psychotropic for agitation |
| | ☐ | Alcohol detox medications (Pabrinex and/or Chlordiazepoxide) |
| **Patient Related** | ☐ | Physical health issues (refer to the yellow box on the right) |
| | ☐ | Patients planned for discharge/leave with outstanding issues requiring follow up |
| | ☐ | New T2/T3 or renewal needed |
| | ☐ | Patient with substance abuse |
| | ☐ | Patient in seclusion |
| | ☐ | Patient on the palliative care pathway |
| | ☐ | Patient with Behavioural and Psychological Symptoms of Dementia (BPSD) when not on a dementia ward |
| | ☐ | Patient who self harm or have suicidal thoughts |
| | ☐ | Patient recently moved from another country (difficult to obtain history, different medications brands) |
| | ☐ | Patient not reviewed within the past 7 days (acute) or fortnight (rehab) by a pharmacist |
| | ☐ | Patient lacking capacity to consent to medication administration |
| | ☐ | Patient with undetermined allergy status |
| | ☐ | Patient <18 years OR >70 years (age) |
| | ☐ | Other …………………………………… |
| | | **Reminder: identifying ≥4 amber criteria for an individual patient will escalate them to a Red risk rating** |

### Green

| | | |
|---|---|---|
| | ☐ | Does not have any Red or Amber criteria |
| | ☐ | Other …………………………………… |

### Outcome

| | | |
|---|---|---|
| 🟥 | ☐ | High risk patient - review every 1 - 2 days |
| 🟨 | ☐ | Moderate risk patient - review every 2-4 days |
| 🟩 | ☐ | Low risk patient - review once every working week or more frequently based on referral |
| | | Comment: |

| | Staff name: | Date: | ☐ Pharmacist. | ☐ Pharmacy technician. |
|---|---|---|---|---|
| ☐ | Staff name: | Date: | ☐ Pharmacist. | ☐ Pharmacy technician. |

### Instruction

This evidence-based tool was developed to assist pharmacy team members in categorising patients based on their risk of developing medicines related problems.

It aims to standardise care, improve patient safety, and optimise pharmacy service delivery by helping pharmacy team members prioritise higher risk patients for pharmacy review.

**Considerations when using the tool:**

- This tool should be reviewed every time a patient is seen by a pharmacy team member.
- The comment section could be used to add more details about the patients, specify required actions, or any other need.
- If, in your professional judgement you disagree with the outcome of the tool, you can tick 'other' in any category stating the reason for this decision.

**Tool development study reference:**

Alshaikhmubarak FQ, Keers RN, Brown P, Lewis PJ. Developing the Inpatient Mental Health Pharmaceutical Assessment and Care Tool (IMPACT) for use by UK mental health pharmacy teams—a modified Delphi study. Br J Clin Pharmacol. 2025; 1-18. doi:10.1002/bcp.70083

### High-Risk Medicines

| | |
|---|---|
| ☐ | Clozapine |
| ☐ | Lithium |
| ☐ | Valproate |
| ☐ | Warfarin |
| ☐ | Direct oral anticoagulant (DOAC) medication |
| ☐ | Insulin |
| ☐ | Zuclopenthixol acetate or Zuclopentixol Acuphase |
| ☐ | Anticonvulsants (e.g. topiramate, levetriacetam) *for seizure |
| ☐ | Strong opioids (e.g. methadone, fentanyl) |
| ☐ | Medications for Parkinson's disease (e.g. levodopa, apomorphine) |
| ☐ | Anti-cancer medications (e.g. azathioprine, fluorouracil) |
| ☐ | Intensive Therapeutic Drug Monitoring (TDM) drugs (e.g. phenytoin, carbamazepine) |
| ☐ | QTc prolonging medication (e.g chlorpromazine, haloperidol, amisulpride) |
| ☐ | Depot antipsychotics |
| ☐ | High Dose Antipsychotic Therapy (above 100% BNF maximum) |

### Physical health issues

| | |
|---|---|
| ☐ | Acute renal impairment |
| ☐ | Acute hepatic impairment (Liver function tests > 3 times upper of limit normal) |
| ☐ | Chronic hepatic impairment |
| ☐ | Chronic kidney Disease Stage ≥ 3b (eGFR < 44ml/min) |

### Physical health issues

| | |
|---|---|
| ☐ | Chronic kidney Disease Stage ≤ 3a (eGFR ≥ 45ml/min) OR electrolytes levels outside reference range |
| ☐ | Moderate hepatic impairment (Liver function tests > upper limit of normal (ULN) but < 3X ULN) |
| ☐ | White blood cells (WBC) OR Haemoglobin levels (CRP, HB1) outside reference range |
| ☐ | Electrolytes levels outside reference range |
| ☐ | Patient with high creatine kinase (CK) |
| ☐ | No VTE assessment for those prescribed antipsychotic |
| ☐ | Outstanding Electrocardiogram (ECG) |
| ☐ | History of fall (For electronic tool: Fall > 1 in the preceding 3 months) |
| ☐ | Patient with physical healthcare issues requiring follow-up by pharmacy team |

**Fig 5. IMPACT tool version 3.**

**Table 7. Examples of the considerations included in the usage manual.**

| Examples of the manual considerations |
| --- |

Reference ranges and medications lists:
- Reference ranges for parameters are expected to be locally added to the tool.
- While examples are included in the tool for QT prolonging medications and medications requiring intensive therapeutic drug monitoring, these are only for guidance. Organisations may have agreed lists of these medications that could be used alongside the tool, yet care must be taken as including all medications, such as all antipsychotics, may risk overclassifying patients as medium-risk when they may be low-risk.

Frequency of review:
- The frequency of review in the tool is for guidance. Each organisation should adapt the frequency based on their local context.
- The initial implementation phase of the IMPACT tool in your organisation should help in determining the appropriate frequency of review. Make sure to consider the variation across different wards and hospitals and involve all pharmacy team members in evaluation and determination of the appropriate frequency of review.

Important considerations:
- Many of the risk indicators are open to interpretation and this was deemed necessary to ensure the flexibility of the tool. Clinical judgement is important and should always be utilised alongside the tool. For example, there is no timeframe for the risk indicator 'patient missed > 2 doses of the same regular prescribed medication'. Clinical judgement should be used here as the timeframe will be different based on the type of medication missed (e.g., depot antipsychotics, antibiotics, etc.).
- If you want to record more details about certain risk indicators, use the comment box. For example, if you tick 'patients with physical healthcare issues requiring follow-up by pharmacy team', you can describe the issues in the comment box such as: 'patient has an acute infection and requires review of antibiotics' or 'patient has increased prolactin' or 'patient has uncontrolled diabetes/ uncontrolled blood pressure'.

acute care patient prioritisation tools in the literature were primarily developed for use by pharmacists only [22–28], these options were not explored before. It was, however, suggested in one study that 'low risk' patients could be reviewed by pharmacy technicians using a modified version suitable for pharmacy technician use [23]. Future work could explore these possibilities in line with the regulations around the changing role of pharmacy technicians.

Ambiguity was another point raised by participants as some risk indicators were thought to be subjective such as 'self-harm'. While this ambiguity was initially viewed negatively by some participants, it was later viewed positively as it was felt to allow more clinical discretion and autonomy when using the tool. Similar opinions were reported in a recent study that observed their acute care pharmaceutical prioritisation tool to be more agreeable by pharmacists compared with other tools, which they partly attributed to the lower specificity of their tool's criteria compared to other tools [29]. The importance of having a room for clinical judgement was also highlighted in another study evaluating a pharmaceutical prioritisation tool for general hospitals [30]. Hence, purposeful ambiguity, which allows for interpretation and clinical judgement, was explained in the usage manual to ensure it is well understood by the tool users.

Training to use the tool may be a critical element in ensuring the effective use of the IMPACT tool [30]. When prepared and delivered effectively, it could overcome some concerns reported by participants in this study such as ambiguity and complexity issues, consequently leading to reduced burden and a more positive affective attitude. Training is also essential for fidelity and sustainability of interventions ensuring appropriate and continuous use of the tool [13]. The training material was modified iteratively in this study to assist future implementation of the IMPACT tool. One example of an important element added to the training was the purpose of the tool. Our findings echoed those of Falconer et al where focus group participants debated whether spending more time reviewing high risk patients was more beneficial than reviewing all patients rapidly [31]. Deprioritising lower risk patients was previously reported to potentially save time and resources [22,29]. The current study suggests a more specific benefit which is allowing more time to be spent on higher risk patients. This highlights the importance of equity

and ensuring all patients receive the level of care they need rather than equally distributing pharmacy staff time between patients. Whilst this may introduce a risk of missing issues that may arise with lower risk patients, the pharmacy team are likely to be notified with changes through other mechanisms such as through the MDT meetings.

While participants generally believed the tool would be useful for improving both patient outcomes and pharmacy service delivery, they acknowledged that further testing was required to ascertain these potential benefits. A suggestion was to conduct a comparison study between current practice in their organisation and the IMPACT tool. Therefore, future research could explore the effectiveness of the IMPACT tool in improving patient outcomes and service delivery using a larger sample size and a mix of qualitative and quantitative approaches. Factors to be assessed may include the occurrence of DRPs, length of hospitalisation, adherence to the tool, and agreement of pharmacy staff with the tool's outcomes.

## Strengths and limitations

This study has several strengths. Firstly, it was multi-centred including perspectives from different specialities across five organisations within England and Scotland. Secondly, the TFA framework which was rigorously developed and commonly used for acceptability testing [17] was used to develop the focus group guide and to structure the analysis. Thirdly, the use of iterative testing allowed making the necessary initial changes to the tool and focusing on other elements in subsequent focus groups. The study, however, was limited by including a single iteration round which may have resulted in an insufficient refinement of the tool and training material. Another limitation was the short duration of data collection as using the tool for a longer period of time on a greater number and range of patients may generate more feedback by participants. Yet, the current duration and single iteration were deemed appropriate as this was planned as an initial small-scale acceptability study. The study was also limited by the low number of participants recruited from one organisation which resulted in a dual interview with two pharmacy technicians. While the presence of a pharmacist may have introduced a different perspective enriching the discussion, this dual interview provided a good opportunity to explore the perspectives of pharmacy technicians without the influence of pharmacists. Power dynamics between focus group participants was previously reported to influence the discussion causing some members to be less likely to speak and express their opinion [32].

## Conclusion

The IMPACT tool was acceptable by participants and insights and recommendations from the focus groups helped further refine and improve it. This was the first attempt to evaluate the IMPACT tool, and future work may further explore qualitative and quantitative feasibility of the IMPACT tool using other implementation frameworks to ensure its efficacy and successful implementation.

## Supporting information

**S1 File. IMPACT tool version 1.**
(DOCX)

**S2 File. Detailed description of participating NHS organisations.**
(DOCX)

**S3 File. Focus group guide.**
(DOCX)

**S4 File. Changes to the training and IMPACT tool version 1, IMPACT tool version 2, and Training session material version 2.**
(DOCX)

**S5 File. The Inpatient Mental Health Pharmaceutical Assessment and Care Tool (IMPACT) – User Manual.**
(DOCX)

**S6 File. IMPACT tool version 3b.**
(DOCX)

## Acknowledgments

The authors would like to express their gratitude to the participating NHS organisations and all participants who generously contributed their time and insights to this study.

## Author contributions

**Conceptualization:** Fatima Q. Alshaikhmubarak, Richard N. Keers, Petra Brown, Penny J. Lewis.

**Data curation:** Fatima Q. Alshaikhmubarak, Richard N. Keers, Penny J. Lewis.

**Formal analysis:** Fatima Q. Alshaikhmubarak.

**Methodology:** Fatima Q. Alshaikhmubarak, Richard N. Keers, Petra Brown, Penny J. Lewis.

**Supervision:** Richard N. Keers, Penny J. Lewis.

**Writing – original draft:** Fatima Q. Alshaikhmubarak.

**Writing – review & editing:** Fatima Q. Alshaikhmubarak, Richard N. Keers, Petra Brown, Penny J. Lewis.

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
