## [Decision Letter · Decision Letter 0]

28 Dec 2025

Dear Dr. Alshaikhmubarak,

Thank you for submitting your manuscript to PLOS ONE. After careful consideration, we feel that it has merit but does not fully meet PLOS ONE’s publication criteria as it currently stands. Therefore, we invite you to submit a revised version of the manuscript that addresses the points raised during the review process.

We look forward to receiving your revised manuscript.

Kind regards,

Yaser Mohammed Al-Worafi

Academic Editor

PLOS One

Journal Requirements:

“This study was part of a PhD funded by the Saudi Arabian Cultural Bureau in London”

3. In the online submission form you indicate that your data is not available for proprietary reasons and have provided a contact point for accessing this data. Please note that your current contact point is a co-author on this manuscript. According to our Data Policy, the contact point must not be an author on the manuscript and must be an institutional contact, ideally not an individual. Please revise your data statement to a non-author institutional point of contact, such as a data access or ethics committee, and send this to us via return email. Please also include contact information for the third party organization, and please include the full citation of where the data can be found.

Reviewers' comments:

Reviewer's Responses to Questions

**Comments to the Author**

1. Is the manuscript technically sound, and do the data support the conclusions?

Reviewer #1: Yes

Reviewer #2: Yes

2. Has the statistical analysis been performed appropriately and rigorously?

Reviewer #1: I Don't Know

Reviewer #2: N/A

3. Have the authors made all data underlying the findings in their manuscript fully available?

Reviewer #1: No

Reviewer #2: Yes

4. Is the manuscript presented in an intelligible fashion and written in standard English?

Reviewer #1: Yes

Reviewer #2: Yes

Reviewer #1: The study assessed the acceptability of the IMPACT tool, designed to help UK mental health pharmacy teams identify high-risk inpatients early. Conducted across five NHS sites, it involved pharmacy staff training, retrospective tool use, and feedback via reflection sheets and focus groups. Overall, the tool was well-received—seen as effective and easy to use—though some pharmacy technicians found certain clinical criteria challenging. Feedback led to improvements in the tool and training materials, resulting in a more user-friendly version and a detailed manual. The study concluded that IMPACT was acceptable and called for further research into its feasibility and impact.

Could the authors clarify the following points please:

1- Line 129: Which two organisations were selected for the initial presentation, and how does this differ from the training session described in lines 136-138? What is the rationale behind selecting these organisations over the others that participated.

2- Line 187: typing error [name].

3- Supplementary file 5/page 6:

a. What strategies could be used to ensure that missing blood tests are reviewed, and the IMPACT tool will be re-visited if results are not immediately available to ensure patient’s safety.

b. Authors refer to how risk indicators are open to interpretations and that clinical judgement could be used to risk stratify patients. Does this apply to pharmacy technicians as well? Would they get any supervision and training required to ensure safe completion and implementation of the tool?

4- Results section:

a. The authors mentioned recruitment commenced 25/07/2024-24/01/2025 (lines 131-132) and the interviews were from October 2024 to Feb 2025. Please check that the dates are consistent.

b. Did all participants complete the study?

Reviewer #2: Overall, the manuscript makes a valuable contribution to the literature on patient prioritization and pharmacy practice in mental health settings. The manuscript is clearly written and methodologically sound. The findings are coherent, well-supported by participant quotations, and offer practical insights relevant to implementation. However, minor clarifications and refinements are required.

1. In Table 3 (Modifications following the first focus group), the original risk indicator “Patients with physical healthcare issues requiring follow-up” was modified to “Patients with physical healthcare issues requiring follow-up by pharmacy team.” Please clarify the rationale for specifying the pharmacy team and explain how risk stratification is influenced by the type of person doing the follow-up required. Specifically, does the risk level differ when follow-up is undertaken by the pharmacy team versus other healthcare professionals (e.g., physicians or nurses)?

2. In Table 3, the combined risk indicator includes “Chronic kidney disease Stage <3b (eGFR 45–59 mL/min)”. This terminology is unclear and not aligned with CKD classification. An eGFR of 45–59 mL/min/1.73 m² corresponds specifically to CKD Stage 3a, rather than “<3b”. I suggest to revise the wording for accuracy and clarity or clearly define the intended stages.

3. Although a formal sample size calculation is not required for qualitative studies, the manuscript would be strengthened by explicitly stating how data saturation (or information power) was considered or judged sufficient across the focus groups and dual interview.

4. Regarding the outcome of the IMPACT tool, I have noticed an overlap between the follow-up intervals for high-risk and moderate-risk patients. Specifically, both categories include a review at every 2 days (high risk: every 1–2 days; moderate risk: every 2–4 days). This overlap may reduce the clarity of risk stratification and follow-up prioritization.

5. In line 187, the statement reads: “Ethical approval was obtained from the University [name] Ethics Committee (19507).” Please specify the full official name of the university and the ethics committee instead of using “[name]”.

6. Please introduce the full term at its first appearance in the Abstract (NHS, line 37) and again at its first appearance in the Introduction (line 68). Thereafter, please use the abbreviation consistently throughout the manuscript.

7. In Table 1, the abbreviation EPMA is used without providing the full term. Please write the full term as a footnote below the table.

**Do you want your identity to be public for this peer review?** For information about this choice, including consent withdrawal, please see our Privacy Policy

Reviewer #1: **Yes:** Dr Gydhia Al-Chalaby

Reviewer #2: No

---

## [Author Response · Author response to Decision Letter 1]

10 Jan 2026

#Reviewer 1

Line 129: Which two organisations were selected for the initial presentation, and how does this differ from the training session described in lines 136-138? What is the rationale behind selecting these organisations over the others that participated.

The presentation described in line 129 was an introductory session about patient prioritisation and the IMPACT tool and its development. Some organisations chose to deliver this presentation to introduce the IMPACT tool to their pharmacy teams, with the aim of raising awareness of patient prioritisation, highlighting its importance, and increasing engagement and participation in the study. At the end of the presentation, the study was briefly introduced in one slide describing what would be involved in participation.

To clarify this, the following sentence was added:

“At two organisations, this was preceded by a presentation to pharmacy team members via Microsoft Teams (MS Teams) to introduce the concept of patient prioritisation, development of the IMPACT tool and study and describe what participation would involve.”

Line 187: typing error [name]. Thanks for pointing this out, it was corrected.

3- Supplementary file 5/page 6:

a. What strategies could be used to ensure that missing blood tests are reviewed, and the IMPACT tool will be re-visited if results are not immediately available to ensure patient’s safety.

b. Authors refer to how risk indicators are open to interpretations and that clinical judgement could be used to risk stratify patients. Does this apply to pharmacy technicians as well? Would they get any supervision and training required to ensure safe completion and implementation of the tool? a. Unresolved clinical issues should be clearly noted in the handover and depending on the importance of the results the staff member completing the tool could highlight the person as ‘red’ until the results are reviewed.

b. This is a very important point. As described in page 10, pharmacy technicians may require additional training that may vary across organisations due to the wide variation in qualifications and experience. The following sentence was added to supplementary file 5/page 6: “The training session should address the use of clinical judgement, clarifying when it is appropriate to use it and when senior advice or support should be sought.”

4- Results section:

a. The authors mentioned recruitment commenced 25/07/2024-24/01/2025 (lines 131-132) and the interviews were from October 2024 to Feb 2025. Please check that the dates are consistent.

b. Did all participants complete the study? A. Thank you for highlighting this point. Recruitment and interviews were two distinct phases of the study. Recruitment took place between 25/07/2024 and 24/01/2025. After recruitment, participants attended the training session, started using the IMPACT tool, and completed the reflection sheets. This created a gap between recruitment and interviews which occurred later, between October 2024 and February 2025.

B. One participant dropped out after the training session as they struggled to find time to participate. This was added in the results section: “A total of 18 pharmacy staff members consented to participate in this study. One pharmacist dropped out from the study as they struggled to find time to participate.”

#Reviewer 2

1. In Table 3 (Modifications following the first focus group), the original risk indicator “Patients with physical healthcare issues requiring follow-up” was modified to “Patients with physical healthcare issues requiring follow-up by pharmacy team.” Please clarify the rationale for specifying the pharmacy team and explain how risk stratification is influenced by the type of person doing the follow-up required. Specifically, does the risk level differ when follow-up is undertaken by the pharmacy team versus other healthcare professionals (e.g., physicians or nurses)?

This was suggested by participants, who explained that the purpose of the indicator is to identify patients at risk due to a lack of timely pharmacy review. They argued that this risk only arises when pharmacy input is specifically required. If a patient instead requires review by another healthcare professional, the absence of frequent pharmacy review does not represent a safety risk attributable to the pharmacy team. The following sentence was added to Table 3’s footnote: “Participants suggested this on the basis that the indicator should only flag risk when timely pharmacy input is required”.

2. In Table 3, the combined risk indicator includes “Chronic kidney disease Stage <3b (eGFR 45–59 mL/min)”. This terminology is unclear and not aligned with CKD classification. An eGFR of 45–59 mL/min/1.73 m² corresponds specifically to CKD Stage 3a, rather than “<3b”. I suggest to revise the wording for accuracy and clarity or clearly define the intended stages.

• Thank you for noting this very important point. This was clarified to: “Chronic kidney Disease Stage < 3a (eGFR > 45ml/min)”

3. Although a formal sample size calculation is not required for qualitative studies, the manuscript would be strengthened by explicitly stating how data saturation (or information power) was considered or judged sufficient across the focus groups and dual interview.

In this study, data collection and analysis occurred concurrently, allowing the research team to assess whether new concepts or perspectives continued to emerge. As data collection progressed, progressively fewer new themes emerged across successive focus groups and the dual interview, suggesting that data saturation had been achieved. The following statement was added to the manuscript: “As data collection progressed, fewer new themes emerged across successive focus groups and the dual interview, with no significant new themes identified in the final data collection, suggesting that data saturation had been achieved.”

4. Regarding the outcome of the IMPACT tool, I have noticed an overlap between the follow-up intervals for high-risk and moderate-risk patients. Specifically, both categories include a review at every 2 days (high risk: every 1–2 days; moderate risk: every 2–4 days). This overlap may reduce the clarity of risk stratification and follow-up prioritization.

This is a very important point. The review intervals were developed and agreed upon during the Delphi questionnaire in our previous study and were intentionally defined as suggested intervals. This approach acknowledges that review frequency is likely to vary substantially between organisations due to differences in staffing levels, workload, and service capacity. For example, staff in one organisation may be able to review high-risk patients daily and moderate-risk patients every two days, whereas staff in another organisation may find the longer end of the suggested intervals (e.g. every two days for high-risk patients and every four days for moderate-risk patients) more realistic. Variation may also occur within the same organisation on different days, for instance due to staff absence or increased service demand. The following sentence was added to supplementary file 1 to explain the overlap in the risk: “Note that the appropriate review frequency will vary between and within organisations due to differences in staffing levels, workload, service capacity, and day-to-day operational pressures, which explains the overlap between the proposed ranges.”

5. In line 187, the statement reads: “Ethical approval was obtained from the University [name] Ethics Committee (19507).” Please specify the full official name of the university and the ethics committee instead of using “[name]”.

This was added.

6. Please introduce the full term at its first appearance in the Abstract (NHS, line 37) and again at its first appearance in the Introduction (line 68). Thereafter, please use the abbreviation consistently throughout the manuscript.

This was added.

7. In Table 1, the abbreviation EPMA is used without providing the full term. Please write the full term as a footnote below the table. Thank you for noting this, the full term was added.

---

## [Editor Report · Decision Letter 1]

12 Jan 2026

Evaluating Acceptability of the Inpatient Mental Health Pharmaceutical Assessment and Care Tool (IMPACT): a multi-site study in the United Kingdom

PONE-D-25-34054R1

Dear Dr. Alshaikhmubarak,

We’re pleased to inform you that your manuscript has been judged scientifically suitable for publication and will be formally accepted for publication once it meets all outstanding technical requirements.

Kind regards,

Yaser Mohammed Al-Worafi

Academic Editor

PLOS One
---

## [Editor Report · Acceptance letter]

PONE-D-25-34054R1

PLOS One

Dear Dr. Alshaikhmubarak,

I'm pleased to inform you that your manuscript has been deemed suitable for publication in PLOS One. Congratulations! Your manuscript is now being handed over to our production team.

Kind regards,

on behalf of

Professor Yaser Mohammed Al-Worafi

Academic Editor

PLOS One